# Endometrial Cancer Arising in Adenomyosis (EC-AIA): A Systematic Review

**DOI:** 10.3390/cancers15041142

**Published:** 2023-02-10

**Authors:** Antonio Raffone, Diego Raimondo, Manuela Maletta, Antonio Travaglino, Federica Renzulli, Daniele Neola, Umberto De Laurentiis, Francesco De Laurentiis, Mohamed Mabrouk, Manuel Maria Ianieri, Renato Seracchioli, Paolo Casadio, Antonio Mollo

**Affiliations:** 1Division of Gynaecology and Human Reproduction Physiopathology, IRCCS Azienda Ospedaliero-Universitaria di Bologna, Bologna, Italy; 2Department of Medical and Surgical Sciences (DIMEC), University of Bologna, Bologna, Italy; 3Pathology Unit, Department of Woman and Child’s Health and Public Health, Fondazione Policlinico Universitario Agostino Gemelli IRCCS, Rome, Italy; 4Gynecology and Obstetrics Unit, Department of Neuroscience, Reproductive Sciences and Dentistry, School of Medicine, University of Naples Federico II, Naples, Italy; 5Obstetrics and Gynecology Unit, Salerno ASL, “Luigi Curto” Hospital, Polla, Salerno, Italy; 6Department of Obstetrics and Gynecology, Faculty of Medicine, University of Cambridge, Cambridge, UK; 7Division of Gynecologic Oncology, Department of Women’s and Children’s Health, Fondazione Policlinico Universitario A. Gemelli IRCCS, Rome, Italy; 8Gynecology and Obstetrics Unit, Department of Medicine, Surgery and Dentistry “Schola Medica Salernitana”, University of Salerno, 84081 Baronissi, Italy

**Keywords:** endometr* malignancy, tumors, tumours, carcinoma, adenomyo*

## Abstract

**Simple Summary:**

A systematic review of the literature was performed to assess the clinicopathological characteristics and survival outcomes of endometrial cancer arising in adenomyosis (EC-AIA). From our analysis, EC-AIA is a rare disease that mainly affects menopausal women and shows symptoms similar to endometrial cancer, but has a challenging preoperative diagnosis. The higher prevalence of the non-endometrioid histotype, advanced FIGO stages, and p53-signature might be behind the worse prognosis of EC-AIA compared to endometrial cancer.

**Abstract:**

Endometrial cancer arising in adenomyosis (EC-AIA) is a rare uterine disease characterized by the malignant transformation of the ectopic endometrium within the adenomyotic foci. Clinicopathological and survival data are mostly limited to case reports and a few cohort studies. We aimed to assess the clinicopathological features and survival outcomes of women with EC-AIA through a systematic review of the literature. Six electronic databases were searched, from 2002 to 2022, for all peer-reviewed studies that reported EC-AIA cases. Thirty-seven EC-AIA patients from 27 case reports and four case series were included in our study. In our analysis, EC-AIA appeared as a rare disease that mainly occurs in menopausal women, shares symptoms with endometrial cancer, and is challenging to diagnose preoperatively. Differently from EC, it shows a higher prevalence of the non-endometrioid histotype, advanced FIGO stages, and p53-signature, which might be responsible for its worse prognosis. Future studies are necessary, to confirm our findings and further investigate this rare condition.

## 1. Introduction

Endometrial cancer (EC) is the most common gynecological cancer in developed countries [1,2,3]. In 22.6% of cases, EC coexists with adenomyosis, a benign gynecologic condition, defined as the migration of glands and stroma from the basal layer of the endometrium to the myometrium [4]. In less than 1% of cases, EC has origins in the malignant transformation of ectopic endometrium within the adenomyotic foci, causing a rare disease known as endometrial carcinoma arising in adenomyosis (EC-AIA) [5,6].

In 1959, Colman and Rosenthal established the criteria for the diagnosis of this disease [5], adapting Sampson’s criteria for diagnosis of ovarian cancer arising in endometriosis to adenomyosis. Thus, according to Colman and Rosenthal’s criteria, EC-AIA was defined by the presence of the following histopathological characteristics: absence of carcinoma in the endometrium or elsewhere in the pelvis; demonstration of carcinoma arising from the epithelium of adenomyosis and not invading from other sites; presence of endometrial stromal cells surrounding the epithelial glands, to support the diagnosis of adenomyosis [5].

Given its rarity, data about the clinicopathological characteristics and survival of EC-AIA are poor and mostly limited to case reports and a few cohort studies [5,7,8].

The aim of this study was to systematically review the literature, to assess the clinicopathological features and survival outcomes of women with EC-AIA.

## 2. Materials and Methods

### 2.1. Study Protocol

This study was performed following an a priori defined protocol. All review stages, including search strategy, study selection, risk of bias assessment, data extraction, and data analysis, were performed independently by 2 authors (M.M., A.R.). In case of disagreement, consensus was achieved through discussion among all authors. Reporting of the whole study followed the PRISMA statement and checklist [9].

### 2.2. Search Strategy

Eligible studies were collected by searching MEDLINE, Web of Sciences, Scopus, ClinicalTrial.gov, Cochrane Library, and Google Scholar from January 2002 to October 2022. Several combinations of the following words were used: endometr*; malignancy; tumour; tumor; neoplas*; cancer; carcinoma; endometrial cancer arising in adenomyosis; EC-AIA; ECAIA. Reference lists of all relevant studies were searched to check for possible eligible studies missed.

### 2.3. Search Selection

All peer-reviewed studies assessing women with EC-AIA were included. We a priori considered the following exclusion criteria: studies with no extractable data, studies with overlapping study populations, review articles, studies reported in languages other than English.

### 2.4. Risk of Bias Assessment

The risk of bias within studies was evaluated using the methodological index for non-randomized studies (MINORS) [10]. In detail, seven domains related to risk of bias were assessed, when applicable, as follows: (1) study aim (had the study a clearly stated aim?); (2) patient inclusion (were all eligible patients included during the study period?); (3) data collection (was data collection performed following an a priori defined protocol?); (4) study endpoints (were the study endpoints appropriate to the study aim?); (5) unbiased study endpoints (were Colman and Rosenthal’s criteria for diagnosis of EC-AIA clearly reported? In particular, Colman and Rosenthal’s criteria were the following: absence of carcinoma in the endometrium or elsewhere in the pelvis; demonstration of carcinoma arising from the epithelium of adenomyosis and not invading from other sites; presence of endometrial stromal cells surrounding the epithelial glands to support the diagnosis of adenomyosis [6]); (6) follow-up (was follow-up at least 24 months?; such a follow-up is considered enough long for women with EC [11]); (7) loss to follow-up less than 5% (were patients lost to follow-up less than 5% of the total sample?). All seven domains were applicable for case series, while only four domains (i.e., domains #1, #4, #5, and #6) were applicable for case reports.

Authors judged each included study for each domain as at “low risk”, “unclear risk”, or “high risk” of bias if data were “reported and adequate”, “not reported”, or “reported but inadequate”, respectively.

### 2.5. Data Extraction and Analysis

Data were extracted from the included studies without modifications. For each study, data extracted were the study country, study design, period of enrollment, patients’ characteristics, EC-AIA histological features, EC-AIA histotype, EC-AIA International Federation of Gynecology and Obstetrics (FIGO) stage, death, recurrence, and expression of immunohistochemical markers.

Kaplan–Meier survival analyses for the risk of recurrence or death were performed and reported graphically via Kaplan–Meier curves; Statistical Package for Social Science (SPSS) 18.0 package (SPSS Inc., Chicago, IL, USA) was used as software for data analysis.

## 3. Results

### 3.1. Study Selection

A total of 431 studies were identified through electronic searches. Forty-one articles remained after duplicate removal and abstract screening, and they were assessed for eligibility (Figure 1). Finally, 31 articles were included in this systematic review [12,13,14,15,16,17,18,19,20,21,22,23,24,25,26,27,28,29,30,31,32,33,34,35,36,37,38,39,40,41,42].

### 3.2. Characteristics of the Included Studies and Study Population

All included studies were case reports [13,14,15,16,17,18,19,20,21,22,23,24,25,26,27,28,29,30,31,32,33,36,37,38,39,40,41], except for four case series [12,34,35,42] (Table 1).

The mean age ± SD was 57.9 ± 9.2 years. Twenty-three (85%) patients were menopausal. The most frequent clinical manifestations were abnormal uterine bleeding (40.5%) and abdominal pelvic pain (27%), while eight (21.6%) patients were asymptomatic (Table 2).

The preoperative diagnosis was uterine sarcoma in 12 (32.4%) cases, atypical myoma in six (16.2%) cases, myoma and/or adenomyoma in eight (21.6%) cases, and EC in two (5.4%) cases. In three (8.1%) cases, no lesions had been detected before surgery (Table 2).

Surgical treatment and staging consisted of hysterectomy in all women, bilateral-salpingoophorectomy in 30 (81%) cases, pelvic lymphadenectomy in eight cases (21.6%), and pelvic and paraortic lymphadenectomy in five (13.5%) cases (Table 2).

Concerning histotype, 20 (57.1%) EC-AIA were endometroid, five (14.3%) clear cell, four (11.4%) serous, four (11.4%) adenosarcoma, one (2.8%) carcinosarcoma, and one (2.8%) mullerian mucinous borderline tumor; histotype was not reported in two cases. Of the endometroid histotype, 10 (58.8%) were grade 1, three (17.6%) grade 2, and four (23.5%) grade 3 (Table 3).

In 29 (82.8%) cases, the endometrium was atrophic, while it showed coexistent foci of adenocarcinoma in four (11.4%) cases, and endometrial hyperplasia in one (2.8%) case. In 13 (35.1%) cases, transition from adenomyotic endometrial epithelium to adenocarcinoma within the myometrium was reported. In three (8.1%) women, EC-AIA arose from cystic adenomyoma (Table 4).

Regarding FIGO stage, 12 (32.4%) patients were stage IA, 10 (27%) stage IB, three (8.1%) stage IC; one (2.7%) stage II, five (13.5%) stage III, and six (16.2%) stage IV. For advanced FIGO stages of EC-AIA, pelvic and/or paraaortic metastases were diagnosed in five (13.6%) cases, while six (16.2%) patients showed metastasis in other sites (Table 3).

In 20 (54%) EC-AIA, immunohistochemical expression of some markers was reported: protein 53 (p53) and estrogens and progesterone receptors (ER and PR) were the most investigated markers. In particular, an abnormal expression of p53 was the most frequent immunohistochemical finding (50%), while expression of ER and PR was inconstant (Table 3).

Twenty-one (80.7%) patients underwent adjuvant therapy after surgical treatment (Table 3).

The mean follow-up time ±SD was 15.4 ± 22 months; fourteen (37.8%) patients were lost to follow-up. Of women with follow-up available, eight (34.8%) reported recurrence: three patients had a primary diagnosis of clear cell EC-AIA, while five had a primary diagnosis of endometroid EC-AIA. Death was reported for one (4.3%) patient, who had undergone total hysterectomy and bilateral salpingo-oophorectomy without adjuvant treatment for a stage IC, clear-cell EC-AIA. The patient developed an inguinal lymph node metastasis 3 months after surgery and died from metastatic disease 60 months after surgery.

We graphically reported disease-free survival using a Kaplan–Meier curve (Figure 2), while we were unable to perform Kaplan-Meier analysis for death, as the event “death” was observed in only one EC-AIA woman, who also showed the longest follow-up time within the whole patient cohort.

### 3.3. Assessment of Risk of Bias among Studies

For the “study aim”, “data collection”, and “loss of follow up” domains (when applicable), all studies were considered at low risk of bias.

For the “patient selection” domain (when applicable), two studies were considered at low risk of bias [12,42], while two studies were judged as having unclear risk of bias [34,35].

For the “study endpoints” domain, the risk of bias was considered low in 15 studies [12,13,14,17,18,20,21,24,29,30,31,32,33,37,41,42], while it was unclear in 16 studies [15,16,19,22,23,25,26,27,28,34,35,36,38,39,40].

For the “unbiased study endpoints” domain, the risk of bias was low in 14 studies [13,14,17,18,20,24,29,30,31,32,33,37,39,40,42], unclear in 11 studies [15,16,19,22,23,26,27,28,34,35], and high in six studies [12,21,25,36,38,41]; in particular, in the studies at high risk of bias, Colman and Rosenthal’s criteria were only partially fulfilled.

For the “follow-up” domain, four studies were at low risk of bias [17,18,29,42], 15 were at unclear risk of bias [15,16,22,23,25,26,27,28,34,35,36,38,39,40], and 12 studies were at high risk of bias [12,13,14,19,20,21,24,30,31,32,33,37,41]; in particular, in studies at high risk of bias, the follow-up duration was less than 24 months.

Results of the risk of bias assessment are graphically shown in Figure 3.

## 4. Discussion

### 4.1. Main Findings and Interpretation

This study showed that EC-AIA is a rare disease that most commonly affects menopausal women and shows symptoms similar to EC, with abnormal uterine bleeding as the most frequent symptom. The preoperative diagnosis can be misleading, while postoperative histological examination shows a lower prevalence of the endometrioid histotype and early FIGO stages than EC; conversely, an abnormal expression of p53 and the need for adjuvant treatment are more common. These findings might explain the worse prognosis of EC-AIA compared to EC.

EC-AIA arises by transformation of the endometrium within adenomyotic foci. Similarly to EC, it can present with different histotypes, such as endometrioid, serous, clear cell, and primary uterine müllerian mucinous borderline tumor [5].

The causes of the neoplastic degeneration of adenomyosis are still unknown. However, adenomyosis and EC share common genetic mutations in the molecular pathways regulating cellular proliferation [4]. Among these, EC and adenomyosis share low expression of mRNA of the Phosphatase and Tensin Homolog (PTEN), mutations in phosphatidylinositol 3-kinase (PI3K)/protein kinase B (AKT)/mammalian target of rapamycin (mTOR), and Catenin Beta 1 (CTNNB1) signaling pathways [43], and loss of heterozygosity in the DNA mismatch repair genes [8].

While EC with coexistent adenomyosis has been investigated and its clinicopathological and survival outcomes assessed [4,44,45,46], only three cohort studies [5,7,8] have been reported on EC-AIA in the last sixty years. In detail, Matsuo et al. and Matchida et al. [7,8] compared cases of EC-AIA reported in the literature to women with EC [7] or EC with coexistent adenomyosis [8] treated in their centers. In both these studies, the authors reported an increased risk of deep myometrial invasion and serous or clear cell histology [7,8] in the EC-AIA group. However, although the presence of EC-AIA was associated with decreased disease-free survival, it was not found to be an independent risk factor for overall survival [7,8]. In a large retrospective observational cohort study by Chao et al. [5], of 2080 patients who underwent surgical treatment for EC, 28 endometroid EC-AIA were identified. When compared to endometroid EC and endometroid EC with coexistent adenomyosis, endometroid EC-AIA showed more favorable histological prognostic factors, such as grade 1, smaller maximum diameter, and less commonly deficient expression of mismatch repair proteins. Moreover, no recurrence or death occurred in these women [5]. Thus, the data from these three studies appeared to be in conflict, with a possible impact from the quality of individual studies in the exploratory analyses by Matsuo et al. and Matchida et al. [7,8] and of patient selection in the retrospective study by Chao et al. In fact, Matsuo et al. and Matchida et al. did not perform a systematic review of the literature, with a risk of bias, within their study evaluation; while Chao et al. exclusively included endometroid histotypes as an inclusion criterion for patient selection [5].

In our study, we systematically reviewed the literature to assess the clinicopathological features and survival outcomes of women with EC-AIA. We found that this rare disease appeared to be similar to EC regarding the age at diagnosis, menopausal status, and symptoms, while differing from it in prevalence of postoperative histological characteristics and prognosis. Moreover, the disease showed a challenging preoperative diagnosis. In fact, the most frequent preoperative diagnoses in women with EC-AIA were uterine sarcoma, myoma/adenomyoma, and atypical myoma, with an EC-AIA diagnosis never being preoperatively suspected. This might be due, on the one hand, to the rarity of the lesion, which has led to a lack of specific features at ultrasound or magnetic resonance imaging, and, on the other hand, to a possible normal endometrium with negative endometrial biopsies at hysteroscopy, as the disease arises in adenomyotic foci. In addition, the symptoms and phenotype also do not seem to help in the preoperative diagnosis, as they appeared to be similar to those of EC and EC with adenomyosis [44]. Indeed, EC-AIA occurred in postmenopausal women and presented with abnormal uterine bleeding, pelvic pain, and vaginal discharge as the main clinical manifestations.

This challenging preoperative diagnosis might have contributed to the lower prevalence of early FIGO stages that we found in women with EC-AIA compared to EC. However, the EC FIGO staging system might need to be adapted for this rare disease: since EC-AIA orginates from the adenomyotic foci into the myometrium, the distinction between FIGO stages IA and IB might be meaningless.

Regarding histology, the most common histotype was well-differentiated endometroid. The hyperestrogenic environment due to adenomyosis might explain the higher frequency of the endometroid histotype among EC-AIA patients [4]. However, the prevalence of such histotype in EC-AIA patients appeared lower than that in women with EC. Indeed, we found that 42% of EC-AIA patients in our study showed non-endometroid histotypes. In particular, similarly to cancer arising from ovarian endometriosis, the clear cell histotype was the most frequent among non-endometroid histotypes. Moreover, the p53-signature also showed a higher prevalence (50%) in women with EC-AIA than that with EC. Since this signature has been related to non-endometrioid histotypes [47], it might explain the higher prevalence of non-endometrioid histotypes in EC-AIA women. In fact, 73% of p53-mutated ECs showed a non-endometrioid histotype, while 42.5% of clear cell ECs showed a p53 mutation [48]. Furthermore, the p53-signature might explain the worse survival outcomes of EC-AIA compared to EC. In detail, in our study, of women with available follow-up data, 34.8% showed recurrence and 4.3% died from metastatic disease.

However, studies investigating the The Cancer Genome ATLAS (TCGA) signature in women with EC-AIA might clarify the independent prognostic value of the molecular signature and the origin of the cancer from the malignant transformation of ectopic endometrium within the adenomyotic foci.

### 4.2. Strengths and Limitations

To the best of our knowledge, this is the first systematic review to assess the clinicopathological features and survival outcomes of women with EC-AIA. In particular, differently from previous reviews and case-control studies [4,6,7], we systematically reviewed the literature and evaluated the risk of bias within studies. Moreover, within this evaluation, our study may be the first study to assess adherence to Colman and Rosenthal’s criteria for diagnosis of EC-AIA. Histologic criteria to identify EC-AIA and separate it from EC with coexistent adenomyosis are crucial and have been debated over the last sixty-years [4]. Colman and Rosenthal established criteria for diagnosis, adding to Sampson’s criteria the presence of endometrial stromal cells into myometrium [49]. In addition, Kumar and Anderson argued for the necessity of demonstrating transitional changes from benign to malignant endometrium within adenomyosis, to allow a stricter diagnosis of EC-AIA [49]. However, since the transition from benign to malignant endometrium is an inconsistent finding [4], we exclusively considered Colman and Rosenthal’s criteria within the risk of bias assessment.

However, our results might have been limited by the small sample size, the study design (mostly case reports), and the overall low quality of the included studies. Nevertheless, our study can provide updated data about a rare disease and direct future studies in the field.

## 5. Conclusions

EC-AIA is a rare disease that mainly occurs in menopausal women, shares symptoms with EC, and is challenging to diagnose preoperatively. Differently from EC, it shows a higher prevalence of the non-endometrioid histotype, advanced FIGO stages, and p53-signature, which might explain its worse prognosis.

Future studies are necessary to confirm our findings and further investigate this rare condition.

## Figures and Tables

**Figure 1 cancers-15-01142-f001:**
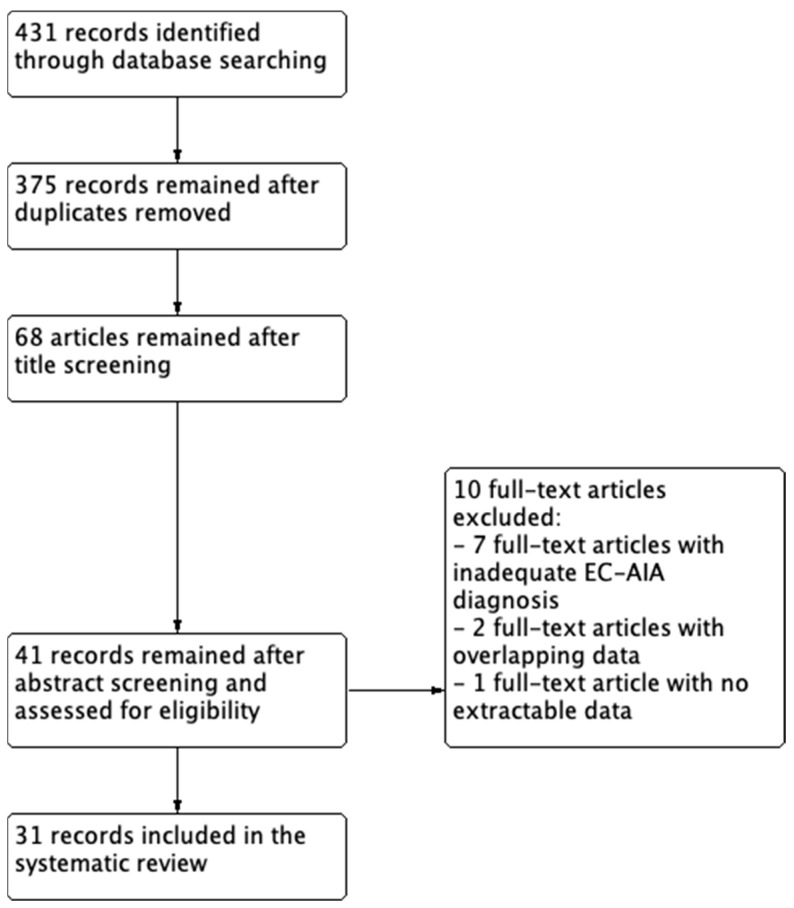
Flow diagram of the studies identified in the systematic review (Prisma template (preferred reporting item for systematic reviews and meta-analyses)).

**Figure 2 cancers-15-01142-f002:**
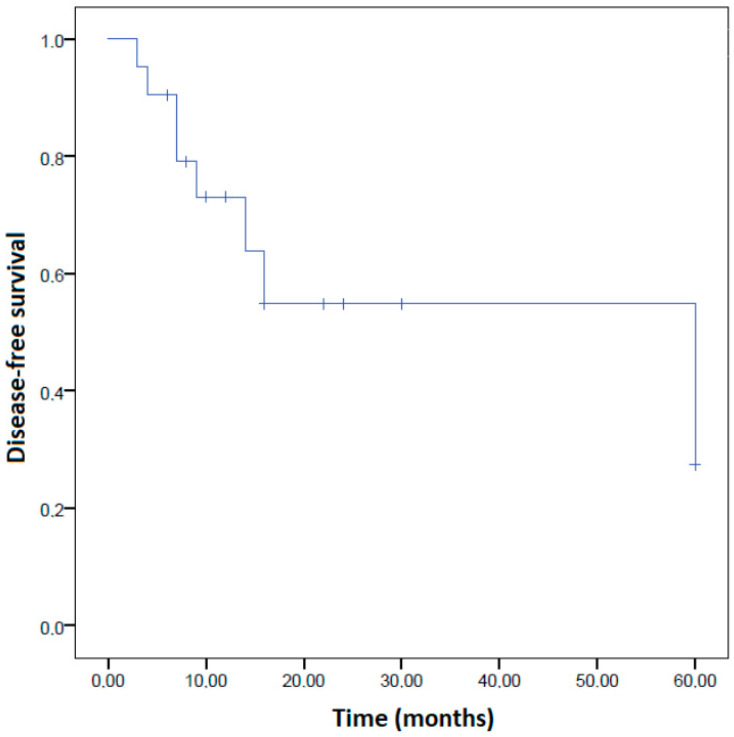
Kaplan–Meier curve for the risk of recurrence in women with endometrial carcinoma arising in adenomyosis (EC-AIA).

**Figure 3 cancers-15-01142-f003:**
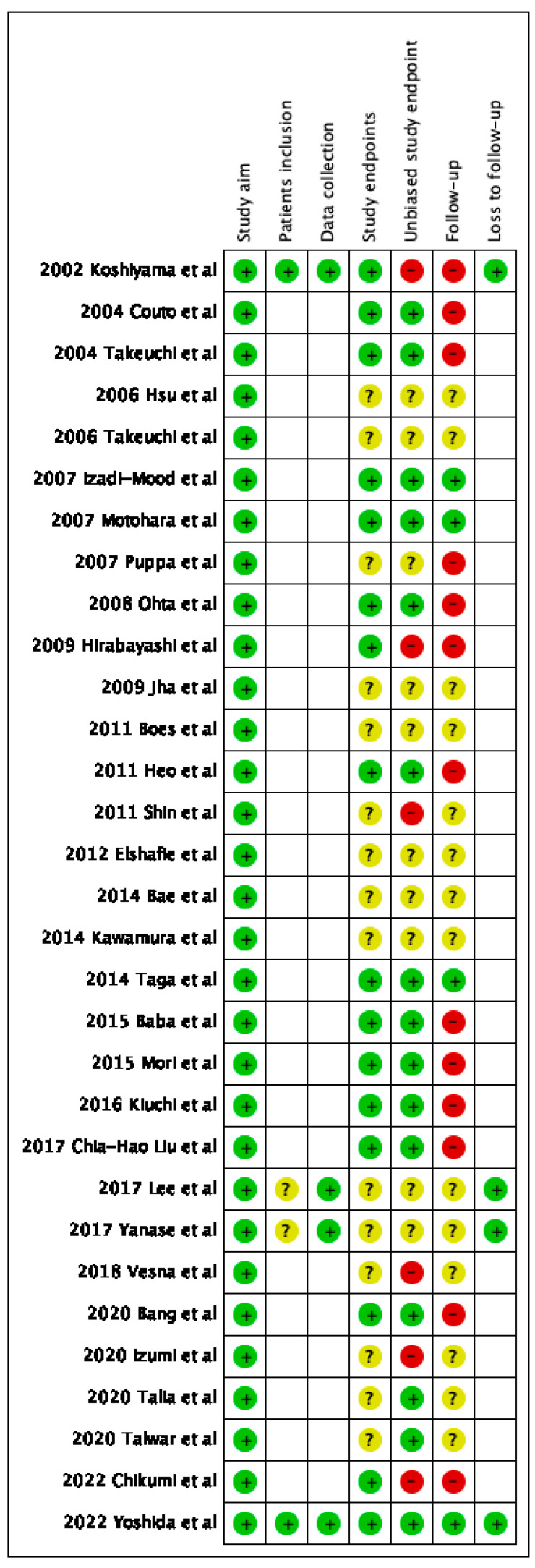
Assessment of the risk of bias. Summary of the risk of bias for each study. Plus sign: low risk of bias; minus sign: high risk of bias; question mark: unclear risk of bias.

**Table 1 cancers-15-01142-t001:** Characteristics of the included studies.

Studies	Country	Study Design	Year
**2002 Koshiyama et al.**	Japan	Case series	1981–2001
**2004 Couto et al.**	Portugal	Case report	2004
**2004 Takeuchi et al.**	Japan	Case report	2004
**2006 Hsu et al.**	Taiwan	Case report	2006
**2006 Takeuchi et al.**	Japan	Case report	2006
**2007 Izadi-Mood et al.**	Iran	Case report	2007
**2007 Motohara et al.**	Japan	Case report	2007
**2007 Puppa et al.**	Italy	Case report	2007
**2008 Ohta et al.**	Japan	Case report	2008
**2009 Hirabayashi et al.**	Japan	Case report	2009
**2009 Jha et al.**	United States of America	Case report	2009
**2011 Boes et al.**	Belgium	Case report	2011
**2011 Heo et al.**	Korea	Case report	2011
**2011 Shin et al.**	Korea	Case report	2011
**2012 Elshafie et al.**	United Kingdom	Case report	2012
**2014 Bae et al.**	Korea	Case report	2014
**2014 Kawamura et al.**	Japan	Case report	2014
**2014 Taga et al.**	Japan	Case report	2014
**2015 Baba et al.**	Japan	Case report	2015
**2015 Mori et al.**	Japan	Case report	2015
**2016 Kiuchi et al.**	Japan	Case report	2016
**2017 Chia-Hao Liu et al.**	Taiwan	Case report	2017
**2017 Lee et al.**	Korea	Case series	2017
**2017 Yanase et al.**	Japan	Case series	2013–2014
**2018 Vesna et al.**	Macedonia	Case report	2018
**2020 Bang et al.**	Korea	Case report	2020
**2020 Izumi et al.**	Japan	Case report	2020
**2020 Talia et al.**	United Kingdom	Case report	2020
**2020 Talwar et al.**	India	Case report	2020
**2022 Chikumi et al.**	Japan	Case report	2022
**2022 Yoshida et al.**	Japan	Case series	2010–2020

A total of 37 patients with EC-AIA were included in our analysis (Table 2).

**Table 2 cancers-15-01142-t002:** Demographic and clinical characteristics of patients with endometrial cancer arising in adenomyosis.

	EC-AIA n (%)
**Number**	37
**Age, (years) mean ± SD**	57.9 ± 9.2
**Parity**	
0	6 (27.3)
≥1	16 (72.7)
missing	15
**Menopause**	
yes	23 (85.1)
no	4 (14.8)
missing	10
**Clinical manifestation**	
Abnormal uterine bleeding	15 (40.5)
Abdominal or pelvic pain	10 (27)
No signs or symptoms	8 (21.6)
Vaginal discharge	1 (2.7)
Others	3 (8.1)
**Preoperative diagnosis**	
Uterine sarcoma	12 (32.4)
Atypical myoma	6 (16.2)
Ovarian cancer	6 (16.2)
Myoma	4 (10.8)
Adenomyosis/adenomyoma	4 (10.8)
None	3 (8.1)
Endometrial cancer	2 (5.4)
**Surgical procedures**	
Bilateral-salpingoophorectomy	30 (81.0)
Pelvic lymphoadenectomy	8 (21.6)
Pelvic and paraortic lymphoadenectomy	5 (13.5)
Other additional surgical procedures	9 (24.3)

Values are given as number (%) unless otherwise noted. **SD** = standard deviation.

**Table 3 cancers-15-01142-t003:** Histological characteristics and survival outcomes of endometrial cancer arising in adenomyosis.

	EC-AIA (%)
**Number**	37
**FIGO stage**	
IA	12 (32.4)
IB	10 (27)
IC	3 (8.1)
II	1 (2.7)
III	5 (13.5)
IV	6 (16.2)
**Histotype**	
Endometroid	20 (57.1)
Clear cell	5 (14.3)
Adenosarcoma	4 (11.4)
Serous	4 (11.4)
Carcinosarcoma	1 (2.8)
Mullerian mucinous borderline tumor	1 (2.8)
Missing	2
Grade	
Grade 1	10 (58.8)
Grade 2	3 (17.6)
Grade 3	4 (23.5)
Missing	3
**Metastasis**	
Pelvic and/or paraaortic nodal metastasis	5 (13.6)
Other sites	6 (16.2)
**Immunohistochemical** **expression**	
P53abn	9 (45)
ER- PR- P53wt	3 (15)
PR+	2 (10)
Others	4 (2)
ER+	1 (5)
ER+ PR+ P53abn	1 (5)
Missing	17
**Adjuvant treatment**	
Chemotherapy and/or radiotherapy	21 (80.7)
Missing	11
Follow-up (months) mean ± SD	15.4 (22)
**Survival Outcomes**	
Death	1 (4.3)
Recurrence	8 (34.8)
Missing	14

Values are given as number (%) unless otherwise noted. **EC-AIA** = endometrial cancer arising in adenomyosis; **FIGO** = International Federation of Gynecology and Obstetrics; **SD** = standard deviation; **ER-** = estrogen receptor absent expression; **PR-** = progesterone receptor absent expression; **ER+** = estrogen receptor expression; **PR+** = progesterone receptor expression; **p53abn** = tumoral protein 53 abnormal expression; **p53wt** = tumoral protein 53 wild type expression.

**Table 4 cancers-15-01142-t004:** Histopathological characteristics of the included studies.

Studies	Histotype	Colman and Rosenthal Criteria *	Endometrium	Transition from Adenomyosis to Endometrial Cancer	Other Histological Features
**2002 Koshiyama et al.**	Endometroid	b,c	Foci of adenocarcinoma	Yes	
**2004 Couto et al.**	Endometroid	a,b,c	Atrophic	Yes	
**2004 Takeuchi et al.**	Endometroid	a,b,c	Atrophic	No	
**2006 Hsu et al.**	Endometroid	a,b,c	Atrophic	Yes	
**2006 Takeuchi et al.**	Endometroid	a,b,c	Atrophic	Yes	Uterine septum
**2007 Izadi-Mood et al.**	Serous	a,b,c	Atrophic	Not reported	
**2007 Motohara et al.**	Endometroid	a,b,c	Atrophic	Yes	
**2007 Puppa et al.**	Endometroid	a,b,c	Atrophic	Not reported	
**2008 Ohta et al.**	Clear cell	a,b,c	Atrophic	Not reported	
**2009 Hirabayashi et al.**	Clear cell	b,c	Foci of adenocarcinoma	Yes	
**2009 Jha et al.**	Mullerian adenosarcoma	a,b,c	Atrophic	Not reported	
**2011 Boes et al.**	Endometroid	a,b,c	Atrophic	Not reported	
**2011 Heo et al.**	Endometroid	a,b,c	Atrophic	Yes	Arising from cystic adenomyosis
**2011 Shin et al.**	Clear cell	b,c	Foci of adenocarcinoma	Yes	
**2012 Elshafie et al.**	Mullerian adenosarcoma	a,b,c	Atrophic	Not reported	Arising in a subserosal adenomyoma
**2014 Bae et al.**	Endometroid	b,c	Not reported	Not reported	
**2014 Kawamura et al.**	Mullerian mucinous borderline tumor	a,b,c	Atrophic	Yes	
**2014 Taga et al.**	Endometroid	a,b,c	Atrophic	Not reported	
**2015 Baba et al.**	Clear cell	b,c	Foci of adenocarcinoma	Yes	Arising from cystic adenomyosis
**2015 Mori et al.**	Endometroid	a,b,c	Atrophic	Not reported	Arising from cystic adenomyosis
**2016 Kiuchi et al.**	Carcinosarcoma	a,b,c	Atrophic	Not reported	
**2017 Chia-Hao Liu et al.**	Serous	a,b,c	Atrophic	Not reported	
**2017 Lee et al.**	Mullerian adenosarcoma	a,b,c	Atrophic	Not reported	
**2017 Yanase et al.**	Not reported	a,b,c	Atrophic	Yes	
**2018 Vesna et al.**	Endometroid	b,c	Atypical Hyperplasia	Not reported	Uterine prolapse
**2020 Bang et al.**	Endometroid	a,b,c	Atrophic	Not reported	Arising in an intramural adenomyoma
**2020 Izumi et al.**	Endometroid	b,c	Not reported	Yes	
**2020 Talia et al.**	Adenosarcoma	a,b,c	Atrophic	Yes	
**2020 Talwar et al.**	Serous	a,b,c	Atrophic	Not reported	
**2022 Chikumi et al.**	Endometroid	b,c	Foci of adenocarcinoma	Not reported	
**2022 Yoshida et al.**	Endometroid	a,b,c	Atrophic	Not reported	

*** Colman and Rosenthal criteria**: a. The cancer must be absent from a normal surrounding endometrium; b. The cancer must be seen to arise from the adenomyotic epithelium, without invasion from another source; c. Endometrial stromal cells must be present in order to support the diagnosis of adenomyosis.

## Data Availability

Not applicable.

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
