# Peer review of "Endometrial Cancer Arising in Adenomyosis (EC-AIA): A Systematic Review"

_cancers, 2023, doi:10.3390/cancers15041142_

Round 1

Reviewer 1 Report

The review paper is written very well, it is clear and concise. The authors explain very well in this paper about uterine disease, endometrial cancer arising in adenomyosis (EC-AIA). I am recommending to publishing this review paper without any modification.  

Author Response

We thank the Reviewer for the kind comments.

Reviewer 2 Report

It is a nice systematic review of the literature in endometrial cancer arising in adenomyosis (EC-AIA).

Author Response

We thank the Reviewer for the kind comment.

Reviewer 3 Report

I have no comment

Author Response

We thank the Reviewer.